DATA RELEASE

# ChestPathCT5-S100: an open real-world chest CT dataset of five common thoracic pathologies with heterogeneous acquisition conditions

Filip Jesionowski[1], David C. Rotzinger[1], Adrien Jayet[1] and Guillaume Fahrni[1,*]

1 Department of Diagnostic and Interventional Radiology, Lausanne University Hospital and University of Lausanne, Rue du Bugnon 46, CH-1011, Lausanne, Switzerland

## ABSTRACT

We present ChestPathCT5-S100, an open dataset of 87 real-world chest CT examinations spanning five common thoracic pathologies: rib fracture, pleural effusion, lung mass, pulmonary embolism, and pneumothorax. The dataset was assembled from a retrospective single-centre cohort over a ten-year period, intentionally preserving acquisition heterogeneity and concomitant findings representative of routine clinical practice. Cases include both contrast-enhanced (arterial and venous phase) and non-contrast examinations, drawn from emergency, oncologic, and trauma settings. All imaging volumes are provided in NIfTI format. Technical validation by two radiologists confirmed correct pathology category assignment, image integrity, and unambiguous visibility of the dominant pathology for each case. A binary co-occurrence matrix of concomitant findings is provided to support multi-label research designs. ChestPathCT5-S100 is publicly available on Zenodo under a CC0 1.0 license, permitting unrestricted use and redistribution. The dataset supports classification, detection, weakly supervised learning, and multi-task learning paradigms.

**Subjects** Imaging, Biomedical Science, Machine learning

Submitted: 23 February 2026

* Corresponding author. E-mail: guillaume.fahrni@chuv.ch

Preprint submitted at https://doi.org/10.20944/preprints202605.1079.v1

## INTRODUCTION

The development of artificial intelligence (AI) systems for medical imaging has become increasingly dependent on the availability of open, well-curated imaging datasets [1, 2]. These datasets form the backbone of algorithm development, enabling reproducible benchmarking and methodological transparency across research groups. Early open medical imaging initiatives were predominantly built around plain radiographs such as CheXpert for chest X-rays [3], whose 2D, single-image nature facilitated large-scale curation. Cross-sectional modalities such as computed tomography (CT) and magnetic resonance imaging (MRI) present considerably greater challenges, owing to their volumetric 3D structure and substantially larger data footprint [4]. While some algorithms have been trained on publicly available online images or textbook illustrations, this approach is limited to 2D tasks and relatively simple models, and cannot be extended to volumetric 3D imaging [5]. Yet despite this recognized need, openly available real-world CT datasets remain persistently scarce [6, 7], creating a structural bottleneck that limits the translation

of medical AI research into clinically meaningful applications. Indeed, access to openly available, representative datasets has been identified as a prerequisite for developing algorithms with genuine clinical utility [8, 9].

Thoracic CT occupies a central role in both emergency and oncologic imaging workflows [10], and serves as the primary modality for detecting a range of clinically consequential pathologies. Among these, pulmonary embolism, pneumothorax, pleural effusion, pulmonary masses, and rib fractures represent frequent and time-sensitive findings that carry significant diagnostic and prognostic weight [11]. The ability to detect and characterize such conditions automatically has motivated a growing body of AI research; however, the practical impact of this work remains constrained by the datasets on which it relies [12, 13].

Synthetic data generation has emerged as an innovative strategy to address the scarcity of real-world imaging data [14]. Nevertheless, synthetically generated datasets carry inherent limitations when compared to genuine clinical data, as they may fail to capture the full spectrum of variability encountered in routine practice, and often underperform models trained on real data [15].

As a step toward addressing this need, we present ChestPathCT5-S100, a curated, fully anonymized, real-world chest CT dataset encompassing five common thoracic pathologies: rib fracture, pleural effusion, lung mass, pulmonary embolism, and pneumothorax. The dataset was assembled from a retrospective single-centre cohort, intentionally preserving the acquisition heterogeneity and concomitant findings characteristic of routine clinical imaging. Released under a CC0 1.0 license and hosted on Zenodo, it is intended to support transparent algorithm development, reproducible benchmarking, and educational use within the medical imaging community. Although the dataset is derived from real clinical imaging, it has been specifically curated and selected to ensure its suitability for research purposes.

## DATA ACQUISITION

ChestPathCT5-S100 was assembled as a retrospective single-centre dataset from an institutional Picture Archiving and Communication System (PACS) over a ten-year period spanning January 2015 to December 2025 (Figure 1). Cases were identified through keyword-based screening of radiology reports, using the conclusion section of the text, followed by systematic image-level confirmation to verify the presence of the target pathology by two radiologists. Only adult patients were included. From an initial pool of 100 cases (20 per pathology category), 13 were excluded due to refusal of research consent or insufficient image quality, yielding a final cohort of 87 cases. The suffix 'S100' in the dataset name reflects the initial approved cohort size of 100 cases, prior to the exclusion of 13 cases during curation. Each case corresponds to a single CT examination from a unique patient. No patient contributed more than one examination to the dataset.

Cases are distributed across five primary pathology categories (Figure 2): rib fracture (RF, $n$ = 14), pleural effusion (Fluid, $n$ = 16), lung mass (Mass, $n$ = 17), pulmonary embolism (PE, $n$ = 20), and pneumothorax (Pneumothorax, $n$ = 20). Each case was assigned to a single pathology category; however, concomitant findings were deliberately preserved and not used as grounds for exclusion, reflecting the complexity encountered in routine clinical practice. The assignment to a primary category was determined by the dominant finding described in the radiologist's report or the primary clinical indication for the exam. In cases



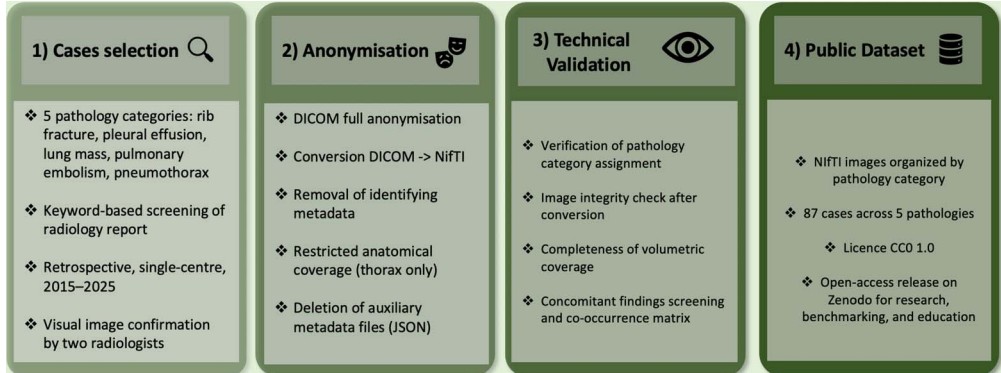

**Figure 1.  Dataset creation pipeline.**
CT cases were selected from a single-centre archive (2015–2025), categorized into five pathology groups, fully anonymised, and converted to NIfTI format. Following technical validation by two radiologists, the resulting dataset of 87 cases is publicly available on Zenodo under a CC0 1.0 license.

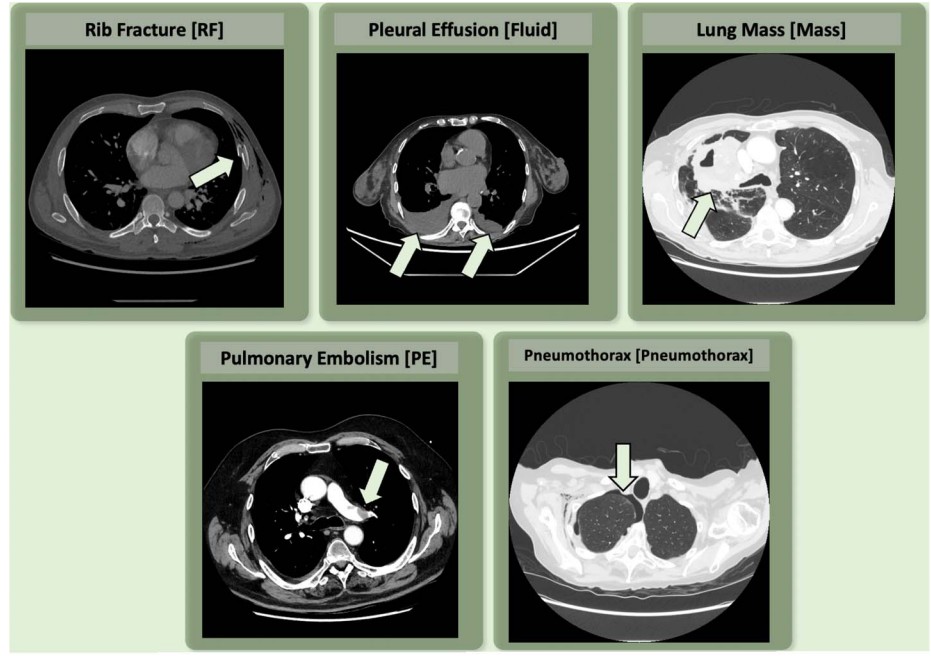

**Figure 2.** Representative axial CT slices (mediastinal, lung, or bone windows) for each of the five pathology categories [Category code in brackets]: pulmonary embolism [PE], pleural effusion [Fluid], lung mass [Mass], pneumothorax [Pneumothorax], and rib fracture [RF]. Light green arrows indicate the primary pathology in each image.

of co-occurrence, the category was assigned based on the finding that first triggered the keyword-based retrieval.

Imaging was performed using CT systems from a single vendor (GE Healthcare, Chicago, IL, USA) in order to limit hardware-related variability while still preserving heterogeneity in acquisition protocols and patient characteristics. Typical acquisition parameters



**Table 1.** Demographic and CT acquisition characteristics of the ChestPathCT5-S100 dataset (*n* = 87).

| Variable | Value |
|---|---|
| **Dataset overview** | |
| Total cases, *n* (%) | 87 (100%) |
| Pleural effusion (Fluid) | 16 (18%) |
| Lung mass (Mass) | 17 (20%) |
| Pulmonary embolism (PE) | 20 (23%) |
| Pneumothorax | 20 (23%) |
| Rib fracture (RF) | 14 (16%) |
| **Demographics** | |
| Age (years), median [IQR] | 64 [46–76] |
| Sex, *n* (%) | |
| Male | 49 (56%) |
| Female | 38 (44%) |
| **CT Protocol, *n* (%)** | |
| Noncontrast CT | 25 (29%) |
| Venous CT | 10 (11%) |
| Arterial CTA | 52 (60%) |
| **CT acquisition parameters** | |
| Matrix size | 512 × 512 |
| Slice thickness (mm), median [IQR] | 0.62 [0.62–1.25] |
| Number of slices, median [IQR] | 481 [264–521] |

IQR, Interquartile Range; PE, pulmonary embolism; RF, rib fracture; CTA, computed tomography angiography.

included a 128 to 256-slice configuration, 0.60 to 2 mm slice thickness and a 512 × 512 reconstruction matrix. Standard injection timing was approximately 40 s for arterial phase acquisitions and 90 s for venous phase acquisitions. For pulmonary embolism studies, acquisition was performed using a bolus-tracking technique (SmartPrep), with scan triggering based on contrast arrival in the pulmonary arteries. The dataset intentionally encompasses heterogeneous acquisition conditions. Both contrast-enhanced (arterial and venous injection time) and non-contrast examinations are represented, and cases were drawn from a variety of clinical indications including emergency, oncologic, and trauma settings. Detailed population characteristics and CT acquisition parameters are reported in Table 1.

## ETHICAL COMPLIANCE AND ANONYMIZATION

The Legal Affairs Unit of CHUV has confirmed that the dataset (project Hors-LHR-BPR1546) falls outside the scope of the Swiss research legislation (Federal Act on Research involving Human Beings, Human Research Act, HRA, SR 810.30) and therefore does not require ethics committee approval, in accordance with Swiss legislation and institutional guidelines.

All cases were fully anonymized prior to release. DICOM metadata were removed using an institutional anonymization pipeline, followed by conversion from DICOM to NIfTI format. Residual metadata embedded in NIfTI headers and associated JSON files was subsequently reviewed and cleared. Imaging volumes were limited to thoracic coverage, ensuring the absence of facial structures that could enable patient re-identification. The anonymization workflow was conducted in compliance with institutional data proion standards at CHUV.



**Table 2.** Pathology co-occurrence matrix. Each column lists cases positive for the given pathology. Cases appearing in more than one column carry concomitant findings. RF, rib fracture; PE, pulmonary embolism.

| RF_pos | Fluid_pos | Mass_pos | PE_pos | Pneumothorax_pos |
|---|---|---|---|---|
| Fluid_013 | RF_001 | RF_019 | Fluid_013 | RF_003 |
| Fluid_018 | RF_002 | RF_009 | | RF_004 |
| Fluid_019 | RF_022 | Fluid_013 | | RF_005 |
| Mass_008 | RF_003 | PE_012 | | RF_006 |
| Mass_015 | RF_006 | Pneumothorax_009 | | RF_007 |
| Mass_018 | RF_007 | | | RF_009 |
| Pneumothorax_003 | Mass_003 | | | Mass_013 |
| Pneumothorax_010 | Mass_008 | | | |
| Pneumothorax_015 | Mass_013 | | | |
| Pneumothorax_022 | Mass_015 | | | |
| | Mass_021 | | | |
| | PE_003 | | | |
| | PE_008 | | | |
| | PE_014 | | | |
| | PE_015 | | | |
| | PE_019 | | | |
| | PE_021 | | | |
| | Pneumothorax_001 | | | |
| | Pneumothorax_003 | | | |
| | Pneumothorax_004 | | | |
| | Pneumothorax_006 | | | |
| | Pneumothorax_010 | | | |
| | Pneumothorax_011 | | | |
| | Pneumothorax_012 | | | |
| | Pneumothorax_013 | | | |
| | Pneumothorax_014 | | | |
| | Pneumothorax_015 | | | |
| | Pneumothorax_019 | | | |
| | Pneumothorax_020 | | | |

The dataset is released under a Creative Commons Zero (CC0 1.0 Universal) public domain dedication, permitting unrestricted use, redistribution, and adaptation without attribution requirements.

## TECHNICAL VALIDATION

Following anonymization and conversion to NIfTI format, all cases underwent systematic review by two radiologists. The cases were divided between the two readers. Verification encompassed correct pathology category assignment, image integrity after DICOM-to-NIfTI conversion, completeness of volumetric coverage, absence of corrupted files, and confirmation that the dominant pathology was unambiguously visible on the imaging volume. No segmentation of pathological structures was performed, and no quantitative annotation validation was carried out.

In addition, each case was screened for the presence of concomitant pathologies beyond the primary category. These findings are reported in Table 2, which provides a binary co-occurrence matrix indicating the presence or absence of each of the five target pathologies across all 87 cases. This matrix enables identification of cases suitable for multi-label research designs.

## REUSE POTENTIAL AND APPLICATIONS

ChestPathCT5-S100 is designed to support a broad range of research and educational applications. The dataset is suitable for AI classification tasks, including both single-label and multi-label paradigms, as well as detection models, weakly supervised learning frameworks, and multi-task learning architectures. The heterogeneous acquisition conditions and deliberate preservation of concomitant findings make it particularly well-suited for evaluating algorithm robustness across varying contrast phases and mixed pathology presentations. Beyond algorithm development, the dataset can serve as a resource for radiology education and pathology training, as well as for testing and benchmarking preprocessing pipelines. The CC0 1.0 license permits unrestricted use, redistribution, and adaptation without attribution requirements.

## LIMITATIONS

Several limitations of this dataset should be acknowledged. The cohort comprises 87 cases, representing a moderate sample size that may limit statistical power in algorithm evaluation studies. As a single-centre dataset, the imaging characteristics reflect a specific institutional acquisition protocol and patient population, which may reduce generalizability to other centres or scanner platforms. No manual segmentations of pathological structures are provided, precluding direct use for supervised segmentation tasks. The dataset contains no clinical ome data or longitudinal follow-up, limiting its applicability to prognostic modelling. A mild class imbalance exists across categories, with rib fracture cases underrepresented relative to pulmonary embolism and pneumothorax. Finally, ChestPathCT5-S100 has not been validated for clinical decision-making and is intended exclusively for research and educational purposes.

## CONCLUSIONS

This work presents an openly accessible and curated dataset of 87 real-world chest CT examinations spanning five common thoracic pathologies, assembled under heterogeneous acquisition conditions representative of routine clinical practice. By preserving concomitant findings and contrast variability, ChestPathCT5-S100 reflects the complexity of real-world imaging more faithfully than synthetic or controlled alternatives. Released under a CC0 1.0 license, it is intended to support reproducible algorithm development, transparent benchmarking, and educational use within the medical imaging community.

## DATA AVAILABILITY

ChestPathCT5-S100 is publicly available on Zenodo [16] and follows a versioned release structure to support reproducibility and future updates. The dataset is organized by pathology category, with each category distributed as an independent compressed archive (.zip). All imaging volumes are provided in NIfTI format (.nii.gz), following the naming convention ChestPathCT5-S100_FC_XXX, where FC denotes the pathology category abbreviation and XXX the case number. Case identifiers are non-consecutive, reflecting the exclusions applied during curation. The total compressed size of the dataset is approximately 10 GB.

## DECLARATIONS

### Ethics approval and consent to participate

The authors declare that ethical approval was not required for this type of research.



## Competing interests

The authors declare that they have no competing interests related to this work.

## Authors' contributions

FJ: Conceptualization, dataset design, data curation, case selection, technical validation, manuscript writing and editing. GF: Conceptualization, project supervision, manuscript review and revision. DCR: Clinical validation, supervision, manuscript review and revision. AJ: Data collection and case selection.

## Funding

Not applicable.

## Acknowledgements

The authors acknowledge the use of artificial intelligence-based writing assistance tools (ChatGPT, OpenAI) for formatting support and language refinement during manuscript preparation.

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
