## [Reviewer Report]

Indicate in the comments box below whether you are happy with the changes made or if the manuscript is unacceptable.Comments on revised manuscriptI am happy with the revision. Thank you and congratulations to the authors.Indicate in the comments box below whether you are happy with the changes made or if the manuscript is unacceptable.Comments on revised manuscriptI am happy with the revision. Thank you and congratulations to the authors.

---

## [Editor Report]

Editor’s AssessmentThe manuscript is ready for formal acceptance.Editor’s AssessmentThe manuscript is ready for formal acceptance.

---

## [Reviewer Report]

Reviewer name and names of any other individual's who aided in reviewer Zekun JiangDo you understand and agree to our policy of having open and named reviews, and having your review included with the published papers. (If no, please inform the editor that you cannot review this manuscript.)YesIs the language of sufficient quality?YesPlease add additional comments on language quality to clarify if needed
Are all data available and do they match the descriptions in the paper? YesAdditional CommentsAre the data and metadata consistent with relevant minimum information or reporting standards? See GigaDB checklists for examples <a href="http://gigadb.org/site/guide" target="_blank">http://gigadb.org/site/guide</a>YesAdditional CommentsIs the data acquisition clear, complete and methodologically sound?YesAdditional CommentsIs there sufficient detail in the methods and data-processing steps to allow reproduction?YesAdditional CommentsIs there sufficient data validation and statistical analyses of data quality? YesAdditional CommentsIs the validation suitable for this type of data?YesAdditional CommentsIs there sufficient information for others to reuse this dataset or integrate it with other data?YesAdditional CommentsAny Additional Overall Comments to the AuthorThis manuscript presents ChestPathCT5-S100, an open dataset of 87 real-world chest CT examinations covering five common thoracic pathologies: rib fracture, pleural effusion, lung mass, pulmonary embolism, and pneumothorax. The authors clearly describe the data source, case selection process, anonymization workflow, data organization, and potential research applications. 1. The authors describe this resource as a “real-world chest CT dataset,” and this wording is generally acceptable because the cases do come from real clinical practice and preserve heterogeneity in contrast-enhanced/non-contrast scans, different clinical settings, and concomitant findings. However, I would suggest adding one sentence in the Introduction or Discussion to clarify that this is a dataset derived from real clinical imaging but curated and selected for research purposes. This would make the use of the term “real-world” more precise. 2. The Zenodo DOI and file naming information should be checked and made fully consistent. The manuscript reports the DOI as 10.5281/zenodo.18256796, whereas the actual Zenodo record appears to be 18256797. 3. There appear to be duplicate references in the bibliography, such as references 14 and 16. 4. Some case identifiers in Table 2 should be checked again to avoid possible spelling or numbering inconsistencies.RecommendationMinor Revision

---

## [Reviewer Report]

Reviewer name and names of any other individual's who aided in reviewer Maurice PradellaDo you understand and agree to our policy of having open and named reviews, and having your review included with the published papers. (If no, please inform the editor that you cannot review this manuscript.)YesIs the language of sufficient quality?YesPlease add additional comments on language quality to clarify if needed
Are all data available and do they match the descriptions in the paper? NoAdditional CommentsI would suggest since you named your project "...S100" to complete the data set in order to have 20 cases for each pathology available.Are the data and metadata consistent with relevant minimum information or reporting standards? See GigaDB checklists for examples <a href="http://gigadb.org/site/guide" target="_blank">http://gigadb.org/site/guide</a>YesAdditional CommentsIs the data acquisition clear, complete and methodologically sound?YesAdditional Comments- please address comment above regarding 100 cases - I am missing information regarding PE-study protocols. - how did you decide in which category you put a study if more than one pathology was present?Is there sufficient detail in the methods and data-processing steps to allow reproduction?YesAdditional Comments- please consider adding labels in the images or segmentations for the pathologies, this would enhance your data setIs there sufficient data validation and statistical analyses of data quality? NoAdditional Comments- is number of cases = number of patients? - if data is normally distributed (I suspect age, slice thickness are not normally distributed), consider reporting median [interquartile range]Is the validation suitable for this type of data?YesAdditional CommentsIs there sufficient information for others to reuse this dataset or integrate it with other data?YesAdditional CommentsAny Additional Overall Comments to the Authordear authors, I think you did a great job putting this together; this cohort could be very helpful for research and education. please consider my comments above, especially to include 100 full cases.RecommendationMajor Revision